# Next-Generation High-Throughput Sequencing to Evaluate Bacterial Communities in Freshwater Ecosystem in Hydroelectric Reservoirs

**DOI:** 10.3390/microorganisms10071398

**Published:** 2022-07-11

**Authors:** Martha Virginia R. Rojas, Diego Peres Alonso, Milena Dropa, Maria Tereza P. Razzolini, Dario Pires de Carvalho, Kaio Augusto Nabas Ribeiro, Paulo Eduardo M. Ribolla, Maria Anice M. Sallum

**Affiliations:** 1Departamento de Epidemiologia, Faculdade de Saúde Pública, Universidade de São Paulo, São Paulo 01246-904, Brazil; martharojas@usp.br (M.V.R.R.); masallum@usp.br (M.A.M.S.); 2FUNDUNESP—Fundação para o Desenvolvimento da UNESP, São Paulo 01009-906, Brazil; 3Instituto de Biotecnologia da UNESP (IBTEC-Campus Botucatu), São Paulo 18607-440, Brazil; p.ribolla@unesp.br; 4Departamento de Saúde Ambiental, Faculdade de Saúde Pública, Universidade de São Paulo, São Paulo 01246-904, Brazil; milenadropa@usp.br (M.D.); razzolini@usp.br (M.T.P.R.); 5Santo Antônio Energia, Porto Velho 76805-812, Brazil; dariocarvalho@santoantonioenergia.com.br (D.P.d.C.); kaioribeiro@santoantonioenergia.com.br (K.A.N.R.)

**Keywords:** bacterial community, metagenomics, 16S, Mansonia, Culicidae

## Abstract

The quality of aquatic ecosystems is a major public health concern. The assessment and management of a freshwater system and the ecological monitoring of microorganisms that are present in it can provide indicators of the environment and water quality to protect human and animal health. with bacteria is. It is a major challenge to monitor the microbiological bacterial contamination status of surface water associated with anthropogenic activities within rivers and freshwater reservoirs. Understanding the composition of aquatic microbial communities can be beneficial for the early detection of pathogens, improving our knowledge of their ecological niches, and characterizing the assemblages of microbiota responsible for the degradation of contaminants and microbial substrates. The present study aimed to characterize the bacterial microbiota of water samples collected alongside the Madeira River and its small tributaries in rural areas near the Santo Antonio Energia hydroelectric power plant (SAE) reservoir in the municipality of Porto Velho, Rondonia state, Western Brazil. An Illumina 16s rRNA metagenomic approach was employed and the physicochemical characteristics of the water sample were assessed. We hypothesized that both water metagenomics and physicochemical parameters would vary across sampling sites. The most abundant genera found in the study were *Acinetobacter*, *Deinococcus*, and *Pseudomonas*. PERMANOVA and ANCOM analysis revealed that collection points sampled at the G4 location presented a significantly different microbiome compared to any other group, with the *Chlamidomonadaceae* family and *Enhydrobacter* genus being significantly more abundant. Our findings support the use of metagenomics to assess water quality standards for the protection of human and animal health in this microgeographic region.

## 1. Introduction

Microorganisms play a significant role in different biomes and provide a wide range of benefits to the ecosystem [1,2]. Ensuring the quality of aquatic ecosystems is a major public health concern to safeguard the health of populations from exposure to pathogens [3]. The assessment and management of a freshwater system can involve the ecological monitoring of the microorganisms present in it, including antibiotic-resistant bacteria, as indicators of both environmental and water quality standards for the protection of human and animal health [4,5]. Additionally, microorganisms affect nutrient cycling and drinking water quality [6,7]. 

Parasites and pathogens are common components of all ecosystems. Microbiological contamination with bacteria associated with anthropogenic activities requires the monitoring of surface water status within rivers and freshwater reservoirs [8]. Human and animal pathogens of enteric origin are important contaminants which can spread throughout the environment via the soil, agriculture activity, water, and sediment [9]. Ecosystem deterioration caused by the global expansion of food production, including crops and animals, can pose a health threat by favoring the emergence of infectious diseases linked to the use of antibiotics, water, pesticides, and fertilizers [10]. These can cause substantial changes in the microbial and pathogen composition of aquatic ecosystems. Sources of fresh water, such as lakes, river basins, and aquifers, are complex interconnected environments. The physicochemical characteristics of a water system affect the diversity and richness of microbial ecosystems containing various organisms [11,12]. Therefore, understanding the composition of aquatic microbial communities can be beneficial for the early detection of water pathogens, improving our knowledge on their ecological niches, and characterizing the assemblages of microbiota responsible for degradation of contaminants and microbial substrates.

Freshwater microbial communities are distinct from those routinely detected in marine and terrestrial ecosystems [13,14]. Studies using the 16S rRNA gene to identify the microbiota present in lakes and reservoirs recovered lineages of *Cyanobacteria* and *Gammaproteobacteria*, mainly when the water was polluted [15]. Much has been learned about the main factors determining lake bacterial communities, such as trophic status, water pH, landscape elements, and retention time [16,17,18]. Freshwater bodies are intimately connected to the surrounding environment and are especially vulnerable to the impacts of increased anthropogenic disturbances, such as land-cover change, urban and rural settlements, and sewage contamination, which can ultimately promote the deterioration of both water quality and ecosystem health. Recent studies have shown that bacterial community diversity and composition are impacted by land use [19,20,21]. A study conducted in a lowland landscape in Central Panama showed that water bacterial community diversity and composition were directly influenced by nearby land use in such a manner that water bodies bordered by forests presented higher diversity and similar community structure, whereas a stream surrounded by a traditional cattle pasture had lower diversity and unique bacterial communities, with a low relative abundance of *Proteobacteria* and *Alphaproteobacteria* and an increased abundance of *Bacteroidetes* [22]. In the Ile-de-France region and northern Germany, changes in land use were associated with fluctuation in the *Cyanobacteria* biomass, which, in turn, was associated with trophic status and water quality [19,20].

In general, the standard methods to identify different toxic components (biological and chemical) in water systems are laborious, time-consuming and require specialized personnel. Thus, the fast assessment of water quality using more sensitive and reliable next-generation sequencing (NGS)-based approaches is needed [23]. The potential of such approaches was demonstrated in a study that evaluated bacterial communities in freshwater in the Amazon River that was facing eutrophication because of changes in the use of soil for crop expansion, animals for meat production, and other anthropogenic changes to natural environments [24].

The tropical rainforest in the Amazon River basin is an extraordinarily diverse ecosystem, comprising thousands of plant and animal species, many endemics to it, with several regions still free from any anthropogenic pressure. However, because the Amazonian landscape is being continuously changed by human presence, this condition may not continue for long [24]. This scenario is likely to have dramatic effects on the flora and fauna interacting with altered bacterial communities present in small lakes, ponds, and water reservoirs. 

One interesting example is the multitude of mosquito species that use the floodplain of rivers as habitats. Different biotic and abiotic elements present in water can influence the selection of oviposition sites by *Aedes aegypti* gravid females [25]. These include the presence of organic matter, surrounding vegetation, moisture, salinity, ammonium, and phosphate [26,27,28,29]. Moreover, microbial communities present in breeding sites have been shown to influence oviposition [30,31]. It has also been well documented that environmental conditions, such as nutritional deficiency, competition, and high temperatures (>30 °C) during larval development can lead to a decrease in the lifespan of mosquito species and increased susceptibility to virus transmission [32]. Additionally, exposure to bacterial microbiota at breeding sites during larval development can affect phenotypic traits related to vectorial capacity, such as egg development, lifespan, and vector competence [33,34].

The results of NGS studies have highlighted the utility of high throughput approaches for identifying pathogens, their taxonomic variation as it relates to water quality, and ecosystem health and sustainability at both local and regional scales. The present study aimed to characterize the bacterial microbiota of water samples collected alongside the Madeira River in rural areas near the Santo Antonio Energia (SAE) reservoir in the municipality of Porto Velho, Rondonia, Western Brazil. A metagenomic approach was used including Illumina NGS of the V3-V4 region of 16s rRNA, and analysis of the physicochemical characteristics of water samples. In the present study, we hypothesized that both metagenomic water quality and water physicochemical parameters would vary across sampling sites in the areas near the Madeira River, including its small tributaries.

## 2. Materials and Methods

### 2.1. Study Site

Fifty-eight water samples were collected along 70 km of the Madeira River (Figure 1, Appendix A) in the municipality of Porto Velho, State of Rondônia, Brazil. The Köppen climate classification subtype for the region studied was AM (tropical monsoon climate-tropical wet climate), with a temperature varying from 21 °C to 34 °C and an annual average temperature of 25.6 °C. The warmest month, on average, is September, with an average temperature of 26.2 °C, and the coolest month is July, with an average temperature of 24.6 °C. The average monthly rainfall varies from a maximum of 264 mm to a minimum of 17 mm. The rainy season is from October to April and the dry season is from June to August, with transition periods in May and September [35]. The locations were chosen because they are in the area of influence of the Santo Antônio Hydroelectric Power Plant reservoir. This is sited close to rural and peri-urban communities inhabited by resettled families due to the water rise of the hydroelectric reservoir, which may impact the water quality in small tributaries and igarapés resulting in changes in the aquatic environment, and, therefore, in the region ecosystem. The collections were undertaken in four different periods: October 2018, February 2019, June 2019 and October 2019. The collection sites represented all the diverse landscapes found in the area of influence of the Santo Antônio reservoir (i.e., the Madeira River floodplain, small tributaries and igarapés) from the Jaci-Paraná district (G1) to the Porto Velho municipality (G5). However, due to restrictions on access to the sites during the dry season, it was not possible to collect an equal quantity of samples throughout the collection periods (Figure 1). 

### 2.2. Sample Collection

To calculate the water parameters, samples were collected directly 30 cm from the surface of the water column, stored in polyethylene bottles, and kept refrigerated until the analysis was performed. The physicochemical parameters used to evaluate the water quality were temperature, hydrogen potential (pH), alkalinity, electrical conductivity, dissolved oxygen (DO), nitrite, nitrate, ammonia nitrogen, total nitrogen, and total phosphorus. Some parameters were measured directly in the field, such as pH, temperature, DO, and electrical conductivity, using a YSI 6920 V2 multiparameter optical probe (YSI, Yellow Springs, OH, USA). The other parameters were analyzed by Venturo Análises Ambientais (https://www.venturoanalises.com.br/ (accessed on 13 May 2022)). For the microbiota study, 1 L of water from each sample site was collected, and a method based on the National Guide for the Collection and Preservation of Samples–Water, Sediment, Communities, Aquatic and Liquid Effluents was used. The water samples were immediately filtered using a sterile filtration unit and vacuum/press pump (115 V/60 Hz; Millipore^®^ Burlington, MA, USA). Each sample was filtered separately using a membrane filter (47-mm diameter, mixed cellulose ester hydrophilic, white, 0.45-μM pore size; Millipore^®^ Burlington, MA, USA). The biomass retained in the membrane filters was individually stored in sterile RNase and DNase-free cryogenic tubes and kept at −20 °C until genomic DNA extraction.

### 2.3. DNA Sample Preparation 

Metagenomic DNA isolation of the microorganisms obtained from the water filtrates was extracted using the Quick-DNA Fungal/Bacterial Miniprep kit (Zymo Research, Irvine, CA, USA) according to the manufacturer’s recommendations. After extraction, DNA was stored at −20 °C until further processing.

### 2.4. DNA Amplification and Sequencing 

Metagenomic DNA was amplified using the V3-V4 region of the bacterial 16S rRNA gene with region-specific primers that included the Illumina flowcell adapter sequences (16S Forward 5′ TCGTCGGCAGCGTCAGATGTGTATAAGAGACAGCCTACGGGNGGCWGCAG and 16S Reverse GTCTCGTGGGCTCGGAGAT GTGTATAAGAGACAGGACTACHVGGGTATCTAATCC) [36]. After amplification, polymerase chain reaction (PCR) products were purified using 0.8× AMPure XP beads (Beckman Coulter, Indianapolis, IN, USA) and indexed using sequencing adapters from the Nextera XT Index kit (Illumina, San Diego, CA, USA). Libraries were normalized and pooled to 1 nM based on quantitative PCR values. Pooled samples were denatured and diluted to a final concentration of 1.4 pM using a 20% PhiX (Illumina) control. Sequencing was performed using a Miniseq Mid Output Kit in the Illumina MiniSeq System (Illumina, San Diego, CA, USA).

### 2.5. Microbiota Analysis 

Microbiota analysis was performed using QIIME2 version 2021.11. Briefly, low quality sequences and chimeras were discarded using the DADA2 pipeline with standard parameters. Alfa rarefaction and beta diversity were performed with a sampling of 4098 reads from each sample. Taxonomy was assigned in QIIME against the GreenGenes ribosomal RNA gene database (GreenGenes 13_8 99% operational taxonomic units [OTUs] from 515F/806R region). Permutational multivariate analysis of variance (PERMANOVA) was performed with QIIME2 based on unweighted UniFrac distances. Briefly, when continuous quantitative parameters were tested using PERMANOVA, average values for each parameter were calculated and divided into the following two groups: one group comprising all values above the average and the other group comprising all values below the average. After pairwise permutations, values with *p* < 0.05 were considered statistically significant. An analysis of compositions of microbiome (ANCOM) test was used to assess differential abundance using the q2-composition QIIME2 plugin.

## 3. Results

This Illumina sequencing produced 1,114,287 high-quality reads that were assigned to OTUs, which corresponded to 4687 taxa, across all water samples collected in the study.

### 3.1. Analysis of Water Physicochemical Properties versus Microbiome Composition

Appendix A shows the results of the physicochemical parameters evaluated to assess the water sample quality. All quantitative parameters listed in the table were individually tested for an association with the microbiome composition found in the collected water samples. After PERMANOVA analysis, only alkalinity was marginally insignificant (*p* = 0.057) compared to microbiome composition in the two different groups (HIGH > 20 and LOW ≤ 20) (Table 1a). The β-diversity metric, represented by the principal coordinate analysis (PCoA), did not indicate any differentiation between the groups (Figure 2). However, ANCOM analysis identified the Mycobacterium genus as significantly more abundant in the group with high alkalinity (Figure 3). 

### 3.2. Geographic Locality versus Microbiome Composition 

Next, the microbiome composition was analyzed, and the water sample results (Appendix A) were divided into the following five distinct groups according to geographic proximity: Group 1 (G1); Group 2 (G2); Group 3 (G3); Group 4 (G4); and Group 5 (G5) (Figure 1). 

Overall, the most abundant genera were Pseudomonas, Deinococcus, and Acinetobacter. However, some taxa were differentially distributed (although not statistically significantly) among the five distinct groups, such as Pseudoxanthomonas, Enterobacteriaceae, Synechococcus, and Erhydrobacter (Figure 4).

The PERMANOVA pairwise analysis revealed that G4 presented a significantly different microbiome compared to any other group (Table 1a). Moreover, ANCOM analysis found the Chlamidomonadaceae family and Enhydrobacter genus to be significantly more abundant in G4 compared to the other groups (Figure 5). PERMANOVA pairwise analysis also showed that G5 microbiome composition was significantly different from G2, but ANCOM analysis did not show any evidence of abundance differentiation. The PCoA results did not provide a robust indicator for grouping regarding the collection locations (Figure 6).

## 4. Discussion

High-throughput sequencing Illumina technology helps to determine microbiota composition more accurately, including non-culturable bacteria, present in freshwater ecosystems. Thus, analyzing microbiota communities in greater detail can provide key information regarding the effect of water quality on public health. In the present study, we hypothesized that both metagenomic water quality and water physicochemical parameters would vary across sampling sites in the areas near the Madeira River, including its small tributaries. We demonstrated that bacterial community diversity was related to the geographic origin, associated with changes in land cover/land use. Moreover, water alkalinity was found, at least to some extent, to be related to microbiota composition. 

Overall, the most abundant genera found in the study were *Acinetobacter*, *Deinococcus*, and *Pseudomonas*. The *Acinetobacter* genus includes both nonpathogenic and pathogenic species that prevail in natural environments, including soils, fresh water, oceans, and hydrocarbon-contaminated sites [37,38]. *Acinetobacter* species show a large amount of metabolic plasticity, and they can process various long-chain dicarboxylic acids and aromatic and hydroxylated aromatic compounds that are associated with plant degradation products [39].

The *Deinococcus* genus is composed of extremophilic non-pathogenic microorganisms that are found in a wide range of extreme habitats, such as deserts, hot springs, cold polar regions, and radiation-contaminated areas, and are resistant to ionizing radiation, UV radiation, desiccation, and hypertonic stress [40,41]. Because of its high stress resistance, this genus can be widely applicable in various fields, such as dealing with soil and water polluted by radiation and heavy metals [41].

The genus *Pseudomonas* consists of a group of medically and biotechnologically important bacteria found in association with plant and animals, and has enormous metabolic versatility [42,43]. *Pseudomonas* are often not abundant within freshwater environments [44], but different *Pseudomonas* species have been isolated from freshwater [45,46], and they can be persistent or transitory members of the freshwater microbiome community [47]. Generally, dispersal of *Pseudomonas* from agricultural to non-agricultural environments may result from the water cycle and stormwater runoff, which may explain the high abundance of this genus found in the Madeira River because heavy rain during flooding cycles throughout the year drains water coming from crop and livestock farms near the river.

Freshwater ecosystems are highly impacted by anthropogenic changes in the environment, mainly because of eutrophication. Agricultural and livestock drainage and runoff can carry a variety of nutrients and agricultural chemicals [48] that can potentially affect bacterial community composition [49,50]. Furthermore, contamination by the herbicide glyphosate stimulates the proliferation and growth of *Cyanobacteria* (e.g., *Planktothrix* spp.), which is the main phylum implicated in the eutrophication process [51].

Some taxa were differentially distributed among the five distinct groups, such as Pseudoxanthomonas, Enterobacteriaceae, Synechococcus, and Erhydrobacte; however, these differences were not statistically significant (Figure 4). Water samples collected in the Group 4 (G4) area presented a significantly different microbiome compared to the other groups that were analyzed. In G4, the *Chlamidomonadaceae* family (a member of the *Cyanobacteria* phylum) was significantly more abundant than in the other groups (Figure 5). Additionally, the *Synechococcus* genus, another member of the *Cyanobacteria* phylum, was found to be more prevalent in G4 (Figure 4). This genus is found worldwide and is related to phytoplankton blooms in freshwater eutrophication [52,53,54]. Remarkably, this genus was also reported to be responsible for a cyanobacterial bloom occurring during a warm, rainy period in the eutrophic hydropower reservoir in Rio de Janeiro, Brazil [55].

The *Enhydrobacter* genus is associated with water eutrophication [56,57], and was also found to be significantly more abundant in G4. However, the *Pseudoxanthomonas* genus, which can remove nitrogen and phosphorous under aerobic conditions and, therefore, has a protective role in eutrophication processes [58,59], was underrepresented in this group. Near the G4 area, extensive areas across the river basin are occupied by cattle farms that can partially explain eutrophication triggered by the livestock drainage onto the floodplain. The runoff from these farming areas is probably contaminating the river downstream, which in turn might explain the slight differentiation in the microbiome composition of G5 compared to G2.

Aquatic bacterial community variations in response to changes in the water physiochemical properties have been reported previously [57,60,61,62]. Environmental drivers of microbial composition may act at a local scale, where the physiochemical properties of the water can cause an enrichment process leading to specific bacteria that are better adapted to the new conditions to become more abundant and dominant. Functional bacterial groups, such as denitrifiers, are largely affected by physiochemical parameters, such as oxygen, temperature, pH, and alkalinity of water [17,57,60,63]. Some of these studies have found that aquatic microbiota composition is associated with hydrology and physiochemical properties of water [64,65,66,67,68]. However, due to the large number of biotic and abiotic factors that can affect community structure in natural systems, it is difficult to precisely identify causal effects, and, often, the hydrological and physiochemical attributes of water courses are not strongly associated with the compositional structure of functional communities [69,70,71]. 

In this study, most water quantitative physiochemical parameters were not statistically associated with microbiota composition found in the water samples that were analyzed. The results of the PERMANOVA showed that the association of alkalinity with microbiota composition was marginally non-significant (Table 1b); however, the *Mycobacterium* genus was found to be significantly more abundant in the high alkalinity group (Figure 3). *Mycobacterium* spp. are important organisms associated with both aquaculture and human diseases and are commonly inhabitants of aquatic environments, including rivers, lakes, ponds, and streams [72]. Usually, high NaCl concentrations inhibit the growth of bacteria in this genus, and its growth rate is enhanced by low concentrations of dissolved oxygen under eutrophic conditions [73,74]. However, there is a scarcity of studies assessing the relationship between *Mycobacterium* spp. growth and water alkalinity.

## 5. Conclusions

A metagenomic approach was used to characterize the bacterial microbiota of water samples collected alongside the Madeira River in rural areas near the SAE reservoir in Porto Velho, Western Brazil. We demonstrated that bacterial community diversity is related to geographic origin, which, in turn, may be associated with changes in land cover/land use. Moreover, water alkalinity was found, at least to some extent, to be related to microbiome composition. Our findings support the use of metagenomic profiling to assess both the environment and water quality standards for the protection of human and animal health in this microgeographic region.

## Figures and Tables

**Figure 1 microorganisms-10-01398-f001:**
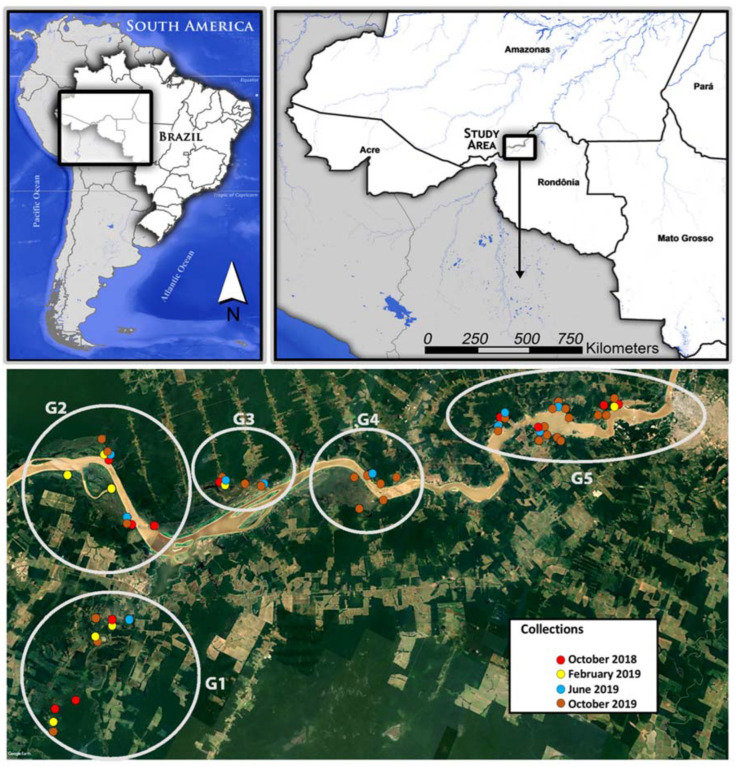
Distribution of the water collection sites along the margins and vicinities of the Madeira River, Porto Velho, State of Rondônia, Brazil. Different colors represent different collection periods, the gray circles highlight five groupings of water samples according to their geographic proximity.

**Figure 2 microorganisms-10-01398-f002:**
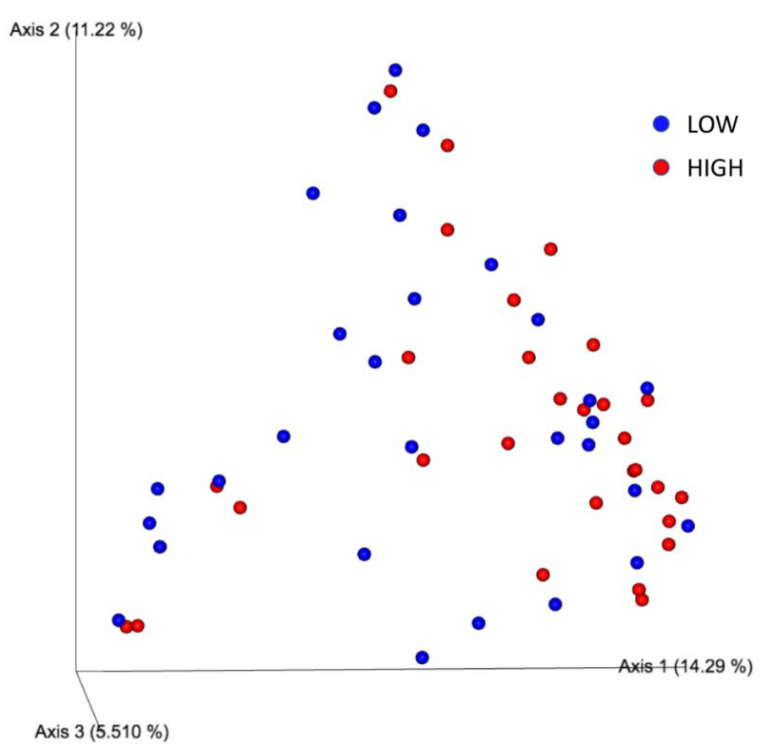
Principal coordinate analysis (PCoA) plot based on unweighted (qualitative) phylogenetic UniFrac distance matrices of water samples according to high or low alkalinity.

**Figure 3 microorganisms-10-01398-f003:**
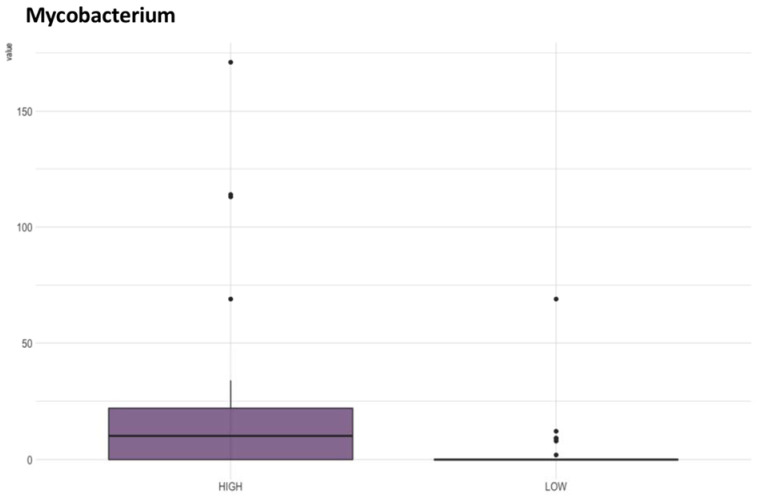
Box plots of ANCOM analysis of water samples according to high or low alkalinity.

**Figure 4 microorganisms-10-01398-f004:**
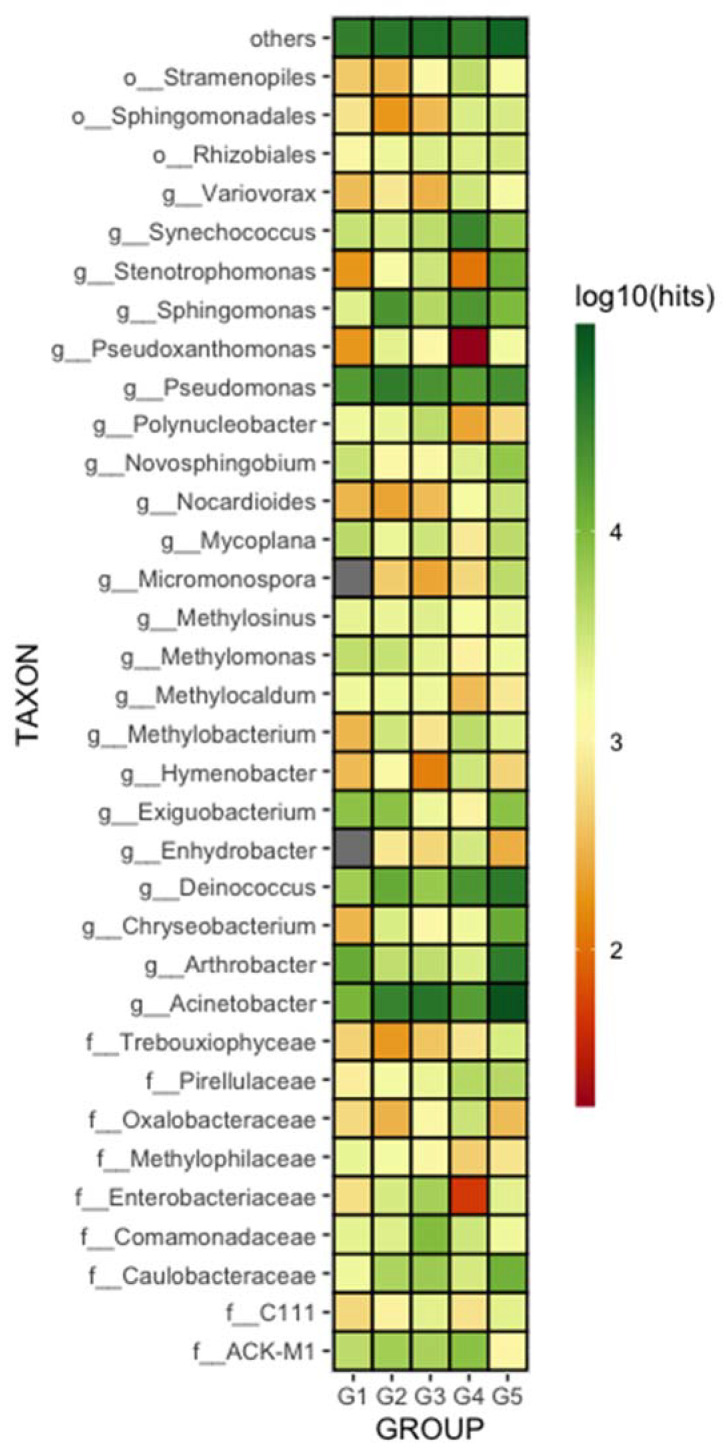
Heatmap of relative abundance of bacterial taxa DNA in water samples according to geographic proximity. Species composition percentages are displayed as the normalized proportion of the microorganism specific counts observed in each sample relative to the total microbial species diversity of the sample (0.5% cut-off). Color gradient key displays the scale of relative abundance in log scale.

**Figure 5 microorganisms-10-01398-f005:**
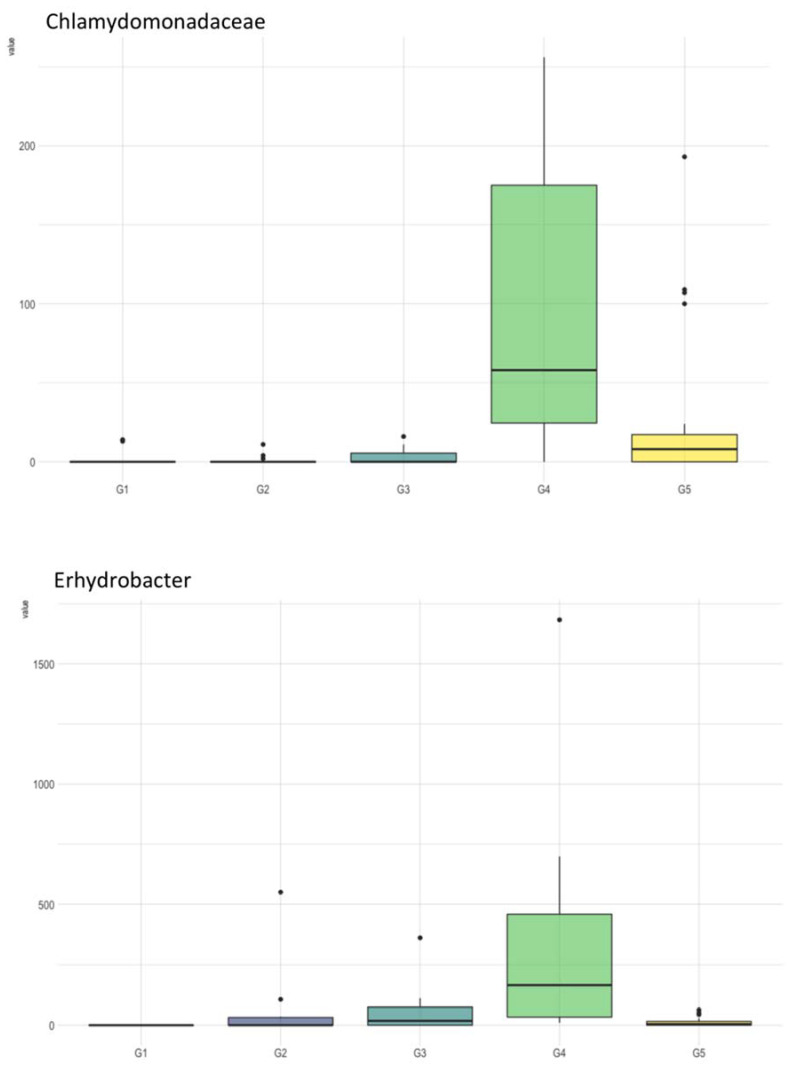
Box plots of ANCOM analysis of water samples according to geographic proximity.

**Figure 6 microorganisms-10-01398-f006:**
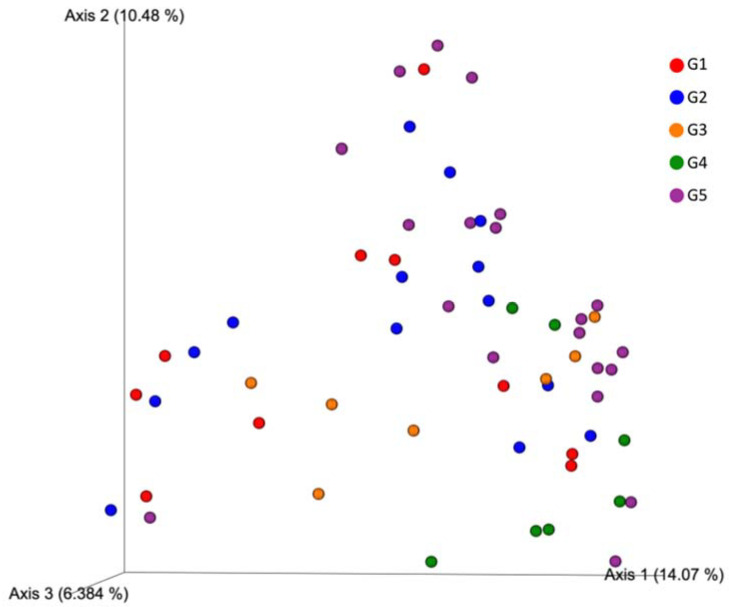
Principal coordinates analysis (PCoA) plot based on unweighted (qualitative) phylogenetic UniFrac distance matrices of water samples according to geographic proximity.

**Table 1 microorganisms-10-01398-t001:** Pairwise PERMANOVA results. (**a**) Group comparisons regarding geographic origin; (**b**) Group comparisons regarding high and low alkalinity. Bold values indicate q < 0.05.

(**a**)								
				**Sample Size**	**Permutations**	**Pseudo-F**	***p*-Value**	**q-Value**
	Group 1		Group 2					
G1		G2		21	999	0.801	0.720	0.720
		G3		17	999	1.243	0.166	0.184
		G4		17	999	2.386	0.002	**0.010**
		G5		33	999	1.686	0.038	0.054
G2		G3		18	999	1.532	0.033	0.054
		G4		18	999	2.992	0.001	**0.010**
		G5		34	999	1.892	0.012	**0.024**
G3		G4		14	999	1.671	0.010	**0.024**
		G5		30	999	1.409	0.068	0.085
G4		G5		30	999	2.084	0.007	**0.023**
(**b**)								
				**Sample size**	**Permutations**	**pseudo-F**	***p*-value**	**q-value**
	Group 1		Group 2					
HIGH		LOW		58	999	1.440455	0.057	0.057

## Data Availability

The datasets generated and analyzed during the current study are deposited in the Sequence Read Archive (SRA) linked to the PRJNA846243 BioProject identifier.

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
