# Peer review of "Next-Generation High-Throughput Sequencing to Evaluate Bacterial Communities in Freshwater Ecosystem in Hydroelectric Reservoirs"

_microorganisms, 2022, doi:10.3390/microorganisms10071398_

Round 1

Reviewer 1 Report

The manuscript describes the metagenomic approach to characterizing the bacterial microbiota of water samples collected by the authors. And, the manuscript describes the diversity of the bacterial community was related to the geographic origin, as well as oxygen, temperature, pH, and alkalinity of water. The manuscript was well-written, and suitable for publication in Microorganisms.

[Suggestions]
The referee suggests that the rationale/research questions/hypothesis could be described both in the Abstract and Introduction (not only in the Discussion and Conclusions).

For example,
"In the present study, we hypothesised that both metagenomic water quality and water physicochemical parameters vary across sampling sites in the areas near the Madeira River, including its small tributaries." (at Discussion, L. 235-238)

"Our findings support the hypothesis of using metagenomic profiling to assess both the environment and the water quality standard for protection of human and animal health in this microgeographic region." (at Conclusions, L. 319-321)

The referee suggests that the main results could be described in the Abstract.

For example,
"The most abundant genera found in the study were Acinetobacter, Deinococcus, and Pseudomonas." (L. 242-243)

Reviewer 2 Report

My comments on the manuscript:

Abstract

There is a lot of theoretical information. Now only the last two sentences relate to the manuscript and even there is only the aim of the research and no results with conclusions. It is necessary to rewrite it and focus on summarizing the content of the manuscript.

Material and Methods

In chapter 2.1. it is necessary to specify and describe the Collection Group. What are their characteristics? Why were these places chosen? What makes them different? It is not enough to mark it in Figure 1 because it is not clear to me and readers from other countries. It is necessary to describe the sampling as such in more detail and add the number of replicates (biological replicates) because now it is confusing. Or are the same numbers within a Collection Site ID as replicates?

Figure 1 – The numbers of grey dots do not correspond to the number of 58 samples listed in Table S1 or to the description of the figure. For example, G3 and G4 both have 7 samples listed in Table S1, 1 and 2 samples in the description of the figure, and three grey dots in both groups in the figure. It is necessary to unify this and state the same numbers everywhere, or to better explain this discrepancy.

Results

Table S2 – It is missing, where relevant, in which units the values for individual measurements are given.

Lines 187-188 – I do not agree with the statement that alkalinity was marginally significant with P = 0.057. In chapter 2.5. you specify your significance limit as P is less than 0.05 and this alkalinity value is still higher. Very little, but still higher. More correctly it should be listed as marginally insignificant. Of course, this does not preclude using the results from this alkalinity in the manuscript.

Line 190 – I do not even see the discrete differences in Figure 2, in my opinion, there are no differences between high and low alkalinity. Maybe the PCoA result in 3D is confusing to me and the differences are better seen in 2D format, ie only using axis 1 and 2. If so, I suggest using a 2D display format.

Table 1 capture – Bold values indicate a q-value higher than 0.05 and not a p-value.

Figure 2 caption – This text corresponds to the caption from Figure 6 and vice versa. This figure illustration is according to high and low alkalinity and not geographic proximity.

Lines 211-212 – This text corresponds to Table 1a and not Table 1b.

Figures 3 and 5 – You need to enlarge the font in these figures. This is unreadable.

Figure 6 caption – This text corresponds to the caption from Figure 2 and vice versa.

Discussion

Line 274 – Delete "Table 1b" because this sentence does not refer to the data in Table 1b at all.

Line 304 – I do not agree with the statement that alkalinity is marginally significant (as I describe above).

However, the whole discussion needs to be rewritten, as it is necessary to link your results with the characteristics of the groups or sites. Why are there abundant bacterial species such as Chlamydomonadaceae and Enhydrobacter in G4? What is characteristic of the G4 group? What is different there than in other groups? It is not enough to describe in the discussion what characterizes the individual types of bacteria you have acquired, but to link your obtained results with other already published results and to explain why these bacteria are in those specific Collection Groups. Yes, in lines 284-288 you partially describe the G4 group, but in more detail, you must describe all groups in this way already in the methodology so that readers already know what the individual groups are like when reading the results.

Round 2

Reviewer 1 Report

OK, the referee agrees with the publication of the current manuscript.

Author Response

Thank you for the acceptance recommendation

Reviewer 2 Report

The authors have edited the manuscript according to my comments, but there are still some inconsistencies:

Line 145 – Check the collection dates about February. Is it February 2018 or 2019? Because in the figure legend you write February 2019 but, in the text, in line 145 you write February 2018.

Line 171 – Delete this text: „The gray dots show the collection sites“ and add the last sentence from the figure caption „Different colors represent different collection periods, the gray circles highlight five groupings of water samples according to their geographic proximity.“

Author Response

Line 145 – Check the collection dates about February. Is it February 2018 or 2019? Because in the figure legend you write February 2019 but, in the text, in line 145 you write February 2018.

Answer: We apologize for the mistake. We corrected the text since the collections were performed during February 2019.

Line 171 – Delete this text: „The gray dots show the collection sites“ and add the last sentence from the figure caption „Different colors represent different collection periods, the gray circles highlight five groupings of water samples according to their geographic proximity.“

Answer: Here again we apologize for the mistake. We corrected the text as suggested.